# Effect of Initial Temperature on the Microstructure and Properties of Stellite-6/Inconel 718 Functional Gradient Materials Formed by Laser Metal Deposition

**DOI:** 10.3390/ma14133609

**Published:** 2021-06-28

**Authors:** Jun Yao, Bo Xin, Yadong Gong, Guang Cheng

**Affiliations:** Department of Mechanical Engineering and Automation, Northeastern University, Shenyang 110819, China; 1870235@stu.neu.edu.cn (J.Y.); gongyd@mail.neu.edu.cn (Y.G.); 1800294@stu.neu.edu.cn (G.C.)

**Keywords:** laser mental deposition (LMD), functionally gradient materials (FGM), preheat, microstructure, tensile properties, microhardness

## Abstract

Stelite-6/Inconel 718 functionally gradient materials (FGM) is a heat-resisting functional gradient material with excellent strength performance under ultra-high temperatures (650–1100 °C) and, thus, has potential application in aeronautic and aerospace engineering such as engine turbine blade. To investigate the effect of initial temperature on the microstructure and properties of laser metal deposition (LMD) functional gradient material (FGM), this paper uses the LMD technique to form Stelite-6/Inconel 718 FGM at two different initial temperatures: room temperature and preheating (300 °C). Analysis of the internal residual stress distribution, elemental distribution, microstructure, tensile properties, and microhardness of 100% Stelite-6 to 100% Inconel 718 FGM formed at different initial temperatures in a 10% gradient. The experimental results prove that the high initial temperature effectively improves the uneven distribution of internal residual stresses. Preheating slows down the solidification time of the melt pool and facilitates the escape of gases and the homogeneous diffusion of elements in the melt pool. In addition, preheating reduces the bonding area between the gradient layers, enhancing the metallurgical bonding properties between the layers and improving the tensile properties. Compared with Stellite-6/Inconel 718 FGM formed at room temperature, the mean yield strength, mean tensile strength, and mean elongation of Stellite-6/Inconel 718 FGM formed at 300 °C are increased by 65.1 Mpa, 97 MPa, and 5.2%. However, the high initial temperature will affect the hardness of the material. The average hardness of Stellite-6/Inconel 718 FGM formed at 300 °C is 26.9 HV (Vickers hardness) lower than that of Stellite-6/Inconel 718 FGM formed at 20 °C.

## 1. Introduction

Stellite-6 is the most widely used wear-resistant cobalt-based alloy and is considered the industry standard for general wear applications, offering excellent resistance to deformation over a wide temperature range and maintaining a reasonable level of hardness at 500 °C. At the same time, it has good impact resistance and corrosion resistance [1]. Inconel 718 is an austenitic nickel-based ultra-high temperature alloy that is widely used in heat resistant parts of aerospace engines and other high temperature resistant components due to its excellent resistance to high temperatures (600–1000 °C), resistance to deformation, and corrosion resistance [2,3,4,5], especially for turbine blades [6,7].

Ni/Co-based functional graded material is an advanced metal matrix composite composed of multi-component alloys and strengthening phases through continuous gradual compounding, which can form hot end parts such as aero-engine turbine blades by laser cladding deposition technology, and has great potential in aerospace applications.

In recent years, as a common additive manufacturing technology, laser metal deposition (LMD) has received extensive attention and research because of its unique advantage in forming FGM parts [8]. Concretely, during the LMD process, several kinds of spherical alloy powders with different properties can be ejected from multiple nozzles simultaneously and then continuously deposited on the substrate through irradiation of high-energy laser beams, as shown in Figure 1. Compared with traditional metal bonding process, LMD process has good metallurgical bonding characteristics and less heat affected zone [9].

Recently, some scholars have conducted extensive research on the forming and performance analysis of Ni/Co-based FGM through LMD in order to explore suitable processing techniques and assess the mechanical properties. Kim et al. [10] used directed energy deposition to deposit Inconel 718 on an AISI1045 substrate and found that the residual stresses during the deposition of lower layers increased abruptly when the substrate was not preheated. When the preheating temperature of the substrate was increased to 200 °C, the fluctuations and differences in residual stresses between successive layers of deposition were significantly reduced, and the mechanical properties were improved. Smoqi et al. [11] used directed energy deposition (DED) to form Stellite 21 on the Inconel 718 substrates and found that preheating the substrate and adjusting the energy density to a moderate level ($E_{V}$ = $\sim 200\;J\cdot mm^{-3}$) were effective in reducing residual stresses, avoiding coating cracks and increasing hardness. Meanwhile, the thermal gradient change in the molten pool with preheating would affect the grain morphology by contributing to the columnar to equiaxed transition. The high preheating temperature can also enhance the diffusion of elements from the precipitated phase into the matrix around it, and that results in the change in microhardness [12].

In addition, the mechanical properties of products obtained using direct laser metal deposition are affected by internal and surface defects such as porosity, unfusion, cracks, and roughness [13], which significantly weaken the mechanical properties of formed structural parts and limit their widespread use [14]. Optimization of LMD parameters can be effective in improving internal defects in Inconel 625/TiB2 gradient structural parts. Numerous studies based on this aspect have been carried out, such as laser power, scanning speed, scanning method, powder defocusing amount, and powder layer thickness [15]. Based on the LMD of Inconel 718 nickel-based alloy, C.Y. Kong et al. developed a low heat input and high-rate deposition causing small and scattered porosity and cracks and successfully prepared many parts using optimized process parameters [16]. Metel et al. increased processing efficiency by installing additional laser beam modulators to control and improve the laser power density distribution, thereby influencing the heat transfer characteristics of the melt pool during processing [17]. Hahn Choo et al. investigated the effect of laser on metallurgical properties in the direct laser metal deposition process and found a linear increase in porosity from 0.13% to 0.88% with decreasing laser power [13].

Abhijit Sadhu et al. used direct laser metal deposition to deposit Inconel 718 and found that stresses due to the thermal gradient between the substrate and the deposited layer at room temperature caused cracks in the fused layer on the substrate, and that preheating the substrate to 300 °C to reduce the cooling rate was effective in alleviating the cracks during multilayer stacking [18,19].

The aim of this paper is to investigate issues such as cracking defects arising from the formation of Stellite-6/Inconel 718 FGMs at room temperature based on the LMD technique. The evolution of the internal residual stress distribution, elemental distribution, microstructure morphology, tensile properties, and microhardness of the formed Stellite-6/Inconel 718 FGM under preheating conditions is investigated to demonstrate the feasibility of preheating the substrate.

## 2. Experimental Condition and Procedure

### 2.1. Experimental Condition and Material

In this paper, the LMD experiments of Stellite-6/Inconel 718 FGM were conducted in a SVW80C-3D (Hybridwise Technology Co., Ltd., Dalian, China) hybrid additive and subtractive machining center with a standard Heidenhain operating system (TNC640, Heidenhain, Berlin, Germany), as shown in Figure 2a. This five-axis linkage vertical machining center is mainly composed of a YLS-2000 fiber laser generator (IPG Photonics Corporation, Oxford, MS, USA), RC-PGF-D-2 double silo negative pressure powder feeder (Zhongke Raycham Laser Technology Co., Ltd., Nanjing, China), ET-80 air compressor (Jaguar Mechanical and Electrical Equipment Co., Ltd., Shenzhen, China), DM-1.5H nitrogen generator (Demiao Technology Co. LTN, Shijiazhuang, China), and data recording system. A 1000 W adjustable temperature-controlled preheater (Taizhou Ouli Electrical Equipment Co., Ltd., Taizhou, China) with a cast copper heating board was adopted to preheat. Nitrogen is selected as protecting and carrier gas to deliver the powder to the substrate through a set of coaxial nozzles. Figure 2b illustrates the principle of LMD process for Stellite-6/Inconel 718 FGM samples.

As for the cladding materials, Stellite-6 cobalt base alloy powder and Inconel 718 nickel base alloy powder with spherical shape were selected to form Ni/Co-based functional graded materials. The particle sizes are in the range of 47–165 μm and 53–150 μm, respectively. Table 1 shows the chemical composition of Stellite-6 and Inconel 718 powders. In order to achieve good metallurgical bonding and avoid cracks in the interface between the first cladding layer and substrate, the forged Stellite-6 cobalt base alloy with a size of 150 mm × 150 mm × 10 mm (length × width × height) was selected as the substrate. Before the experiment, the surface of the substrate was polished with 400 mesh coarse sandpaper to remove the oxide scale, impurities, and dirt, and then, the substrate was fixed on the rotary table of the machine center with the heat resisting tooling fixture. The surface of the substrate after sandpaper grinding was kept level to ensure an uniform during the laser cladding process.

### 2.2. Experimental Parameter and Material

According to previous studies, overlap rate and scanning speed play an important role in the LMD process, and the microstructure and properties of the formed thin-walled parts change with the width of the thin wall [18]. In this study, the process parameters of LMD were set as follows: laser power P was 1000 W, laser scanning rate V was 360 mm/min, and powder feeding rate was 13.5 g/min. In addition, the distance from the laser beam focus to the molten pool is: the defocus D is 13.5 mm. The composition of powder material was selected from 100% Stellite-6 at the bottom to 100% Inconel 718 at the top with a linear change of 10% as a step along the gradient direction. The height of the thin-wall sample was 41.5 mm with a total of 33 layers, and each gradient region consisted of 3 layers. Ten groups of functional gradient material samples were formed. The distribution of the Stellite-6/Inconel 718 FGM sample is shown in Figure 3.

In the experiment, the elements diffusion, microstructure morphology, tensile properties, and microhardness of the FGM thin-walled parts formed at room temperature and 300 °C were tested. The metallographic sample surface was obtained by cutting, grinding, polishing, and etching (40 mL HCl+ 20 mL HNO_3_). Dandong Haoyuan DST-17 X-ray stress analyzer (Haoyuan Technology Co., Ltd., Dandong, China) was used to detect the residual stress of the formed FGM parts. VHX-1000E super depth of field microscope (Keyence Co., Ltd. Beijing, China) was used to observe the macroscopic morphology and microstructure. An OLS4100 3D microscope (Olympus Corporation, Tokyo, Japan) and a scanning electron microscope (SEM) with ZEISS ULTRA PLUS (Carl Zeiss AG, Oberkochen, Germany) were selected to observe the fracture morphology of tensile specimens. The chemical composition of gradient layers elements was analyzed by an energy spectrometer with SEM. Microhardness measurements were conducted on a HVS-1000M Vickers Hardness Tester (LEDI Instruments Co., Ltd., Ningbo, China) (test pressure 1 kg, loading time 10 s).

## 3. Result and Discussion

### 3.1. Micro Morphology

The initial temperature will dramatically affect the forming quality of LMD parts. Ni/Co-based high temperature alloys have a high crack sensitivity and are highly susceptible to cracking through LMD forming of FGMs [20]. As can be seen from the macroscopic morphology of FGM thin-walled samples formed at initial room temperature and preheating temperature (300 °C), a large crack exists at the bottom boundary of FGM thin-walled parts in Figure 4a, and the crack area is enlarged as shown in Figure 4c. The main reason is that a high thermal gradient between the molten pool and the substrate at the beginning of cladding will produce the high thermal stress. With the increase in the cladding layers, the temperature gradient between the adjacent layers decreases gradually. Specially, the thermal stress tends to appear between the adjacent gradient areas (measurement point 1 and 2 in Figure 4b). As the sample is formed and cooled, the large residual stress caused by the different solidification characteristics of the gradient alloy is produced inevitably and leads to poor metallurgical bonding.

Aiming at the above problems, five sets of FGM samples were formed under preheating conditions. Figure 4d shows that no obvious macroscopic cracks appeared in a thin-walled sample. The experimental result is similar to the result of the literature [19] that preheating the AISI-SABE 4340 steel to 1695 K before laser deposition can effectively prevent the generation of macroscopic cracks. Similarly, Lestan et al. [21] verified that the number of cracks in the LMD coating was significantly reduced by preheating the substrate to 360–400 W during the FGM deposition of three types of powders (Metco 15 E, Colmony 88, and Vim Cru 20) on the cast iron.

In order to explore the distribution of residual stress in thin-walled parts, six points were selected at equal spacing (7 mm) between the edge and the middle of the FGM thin-walled parts for residual stress detection. The results of the measured surface residual stress were fitted and are shown in Figure 5a. The mean residual stress at the edge of the FGM samples formed at room temperature changed from tensile stress (138 MPa) near the substrate to compressive stress (−136 MPa) and, then, increased linearly to maximum tensile stress (385 MPa). The maximum difference of mean residual stress is up to 168 MPa at the edge of the samples (near the measured point 3), which demonstrates that the residual stress distribution is far from uniform and more cracks would be produced [19]. In the middle position, the mean residual tensile stress decreases from 124 MPa to 67 MPa and, then, increases to 434 MPa. Figure 5b shows the difference of residual stresses at different heights in the edge and the middle regions of the sample at different initial temperatures. The preheating of the substrate reduces the temperature gradient between the substrate and the cladding layer, effectively improving the uneven distribution of residual stresses within the Stellite-6/Inconel 718 FGM samples and avoiding the formation of cracks.

### 3.2. Element Distribution

The Stelite-6/Inconel 718 FGM element distribution design is shown in Figure 6, with a uniform linear distribution of each element between the gradient layers. To investigate the elemental distribution between the different gradient layers of the LMD FGM samples at room temperature and 300 °C. The SEM results are shown in Figure 7a,b. In order to find the test points conveniently, the center point of each gradient layer along the deposition direction on the metallographic sample is selected as the mark. As can be seen from Figure 7, with the increase in the proportion of Inconel 718, the volume percent of Ni and Fe increased, while the contents of Co and Cr decreased. The overall element distribution trend at room temperature and 300 °C was roughly the same. In order to explore the influence of initial temperature on the element distribution in Stellite-6/Inconel 718 FGM, the measured contents of Ni, Fe, Co, and Cr in the gradient zone were estimated by using linear fitting method. Concretely, the correlation coefficient *R*^2^ is calculated to evaluate the degree of correlation between the actual measured value, and the fitting curve and the results show that when the initial temperature increases from room temperature to 300 °C, $R_{Ni}^{2}$ increases from 0.9391 to 0.9829, $R_{Co}^{2}$ increases from 0.9775 to 0.9912, $R_{Fe}^{2}$ increases from 0.9239 to 0.9388, and $R_{Cr}^{2}$ increases from 0.9546 to 0.9736. The elements show good linear relationship, and the element diffusion and fusion between layers are more uniform. Comparative analysis: preheating will delay the solidification of the melt pool and promote a homogeneous transition of the elements between the gradient layers [22], which is conducive to avoiding the sudden change in interlayer properties.

### 3.3. Microstructure

In this section, the effects of different initial temperatures on the dendrite morphology of the Stellite-6/Inconel 718 FGM thin-walled parts are studied. According to the available literature [23], the stability of the solid–liquid interface depends on the solidification rate (R) and local temperature gradient (G) during the LMD process. Moreover, the G/R ratio determines the transition from columnar to equiaxed grains. Figure 8a–c shows the bottom, middle, and upper microstructure morphology along the longitudinal section of the FGM thin-walled parts at room temperature. Based on the literature [24,25], solid–solid heat transfer from the molten pool of the cladding layer to the substrate is the main way of creating thermal conduction, resulting in rapid heat dissipation and cooling. More columnar grains are generated at the bottom of the thin-walled part and grow epitaxial perpendicular to the lower boundary of the molten pool. The crystal orientation is consistent with the direction of the partial temperature gradient. Only a small number of large equiaxed grains exist below the columnar grains. In Figure 8b, with the continuous accumulation of cladding layers, the heat dissipation mode gradually transforms to heat radiation and air convection. According to the finding of the literature [25], because of the similar crystal structure and chemical composition of the metal materials in the gradient interlayer, the dendrite growth direction (black arrow direction in Figure 8a,b) between the sedimentary layers has the same general orientation, which is at a certain angle to the scanning direction (white arrow direction in Figure 8).

Due to the influence of temperature, the grain growth direction of the latter cladding layer does not inherit the growth direction of the former solidified layer completely during the deposition process, resulting in an obvious bonding area (the area between the two white lines) between the layers. At the top of the thin-walled parts, the heat dissipation methods are mainly heat radiation and air convection. As the heat dissipation rate of air is slower than that of solid, the temperature gradient between the top cladding layers is smaller, and the transition from columnar grains to equiaxed grains occurs at the top, as shown in Figure 8c.

By comparison, as shown in Figure 8d–f, the microstructure morphology in the same locations of the FGM thin-walled parts under preheating condition (300 °C) was obtained. In Figure 8d, preheating the substrate reduces the thermal gradient between the substrate and the cladding layer. Thus, the growth of columnar grains was restrained, and coarser equiaxed grains exist at the bottom. In Figure 8e, similar to Figure 8b, the change in heat dissipation mode makes a disordered crystallization region generate. Due to the reduction in temperature gradient, the bonding region between the adjacent cladding layers becomes narrower, which increases the metallurgical bonding properties of the FGM. In Figure 8f, the heat dissipation mode is thermal radiation and air convection heat dissipation. The narrowing of temperature gradient is more conducive to the nucleation and growth of equiaxed grains, and a large number of more refined equiaxed grain regions appear at the top.

### 3.4. Tensile Properties

In order to investigate the effect of initial temperature on the mechanical properties of the LMD Stellite-6/Inconel 718 FGM, tensile strength and microhardness were tested, respectively. In this subsection, three gradient tensile specimens were prepared by LMD and wire electrical discharge machining (WEDM) processes for exploring the tensile properties of the FGM thin-walled parts. The geometry size, position, and fracture locations of these tensile samples are shown in Figure 9 and Figure 10. The experimental results of the stress-strain curves and the yield strength (YS), tensile strength (TS), and elongation (EL) of different materials are listed in Figure 11 and Table 2, respectively.

Based on Figure 11, the tensile properties of FGM thin-walled parts formed at room temperature are better than Inconel 718 single material along the gradient direction. the mean yield strength, mean tensile strength, and mean elongation along the gradient direction of the thin-walled Stellite-6/Inconel 718 FGM parts formed at room temperature are 44.8 MPa, 84.8 MPa, and 2.55% higher than the Inconel 718 sample. Moreover, the mean yield strength, mean tensile strength, and mean elongation of the FGM tensile specimens in 300 °C preheating condition are 65.1 MPa, 97 MPa, and 5.2% higher than those specimens formed at room temperature. Therefore, the gradient thin-walled parts formed by LMD at 300 °C effectively improve the mechanical properties compared to room temperature.

As can be seen from Figure 10, the fracture location is at the gradient layers with the material proportion of Stellite-6 (90%) + Inconel 718 (10%). Figure 12 and Figure 13 shows the SEM analysis results of fracture morphology of the FGM tensile specimens. The fracture morphology of the FGM tensile specimens formed at room temperature was observed under 20, 200, and 1000 magnification. Based on Figure 12b, the fracture morphology of the FGM tensile specimens formed at room temperature shows a large number of disordered and different shapes of dimples as observed under 200 magnification. Some of the dimples are small and shallow, with strong crack expansion and poor plasticity. The microscopic fracture morphology characteristics reflect the ductile fracture mechanism of the specimen, because the existing dimples indicate that the specimen has a certain degree of elongation deformation before fracture. There are several second phase particles (alloy compounds) with various shapes in the dimples, which are in the shape of fragments. Part of the fracture was observed to be flat and smooth, and the surface was obviously torn, indicating that the fracture occurred in the tensile specimen belongs to brittle fracture. The brittle element Si at the location of the brittle fracture (P1) was up to 0.96 wt.% (Figure 12d). Therefore, the concentration of brittle elements reduces the strength of Stellite-6/Inconel 718 FGM.

Furthermore, a large number of pores with various sizes could be observed at the fracture in Figure 12a and Figure 13a. Image J software was used for statistical calculation of the pore area. As the substrate was preheated, the proportion of pore area decreases from 3.77% (room temperature) to 2.37% (300 °C). It can be explained that preheating can slow down the solidification time of the molten pool, which facilitates the emission of the residual gas such as oxygen and nitrogen in the molten pool. A lower porosity significantly contributes to the mechanical performance of the LMD FGM parts.

In addition, the fracture morphology under a high magnification microscope was observed in Figure 13b. It can be seen that some dimples were evenly and densely distributed at the fracture. According to Zhao et al. [26], the small and deep dimples are conducive to the precipitation of the second phase particles. The second phase particles can enhance the strength and toughness of the FGM tensile specimens because these particles play the role of fine grain strengthening, hindering of grain growth and crack propagation. The results of energy spectrum analysis of the second phase particles show that the content of carbon and oxygen elements increases obviously, indicating that the carbon and oxygen elements in the alloying elements react with other elements in the alloying powder to form carbides and oxides.

### 3.5. Microhardness

Figure 14 shows the distribution of the FGM longitudinal section microhardness test points along the gradient direction (red arrow in Figure 14a). The center position of the gradient layer was selected as the test point, and the hardness values of each gradient were measured three times to determine the average value. The measurement results were shown in Figure 14b. The mean value of maximum hardness of FGM thin-walled parts formed at room temperature is 413.1 HV at 100% Stellite-6. The mean value of minimum hardness is 229.1 HV at 100% Inconel 718, and the average hardness is 315.3 HV. Compared with FGM formed at room temperature, the maximum, minimum, and average hardness values of FGM thin-walled parts formed at 300 °C decreased obviously. The above results are due to the cooling rate is one of the important factors to determine the hardness of thin-walled parts [27]. Preheating can prolong the solidification time of a molten pool by slowing down the heat dissipation rate and reducing the temperature gradient between the gradient layers. This is the same as found by Andreas Weisheit et al. [12]. The hardness of FGM formed at two different temperatures is similar, and the hardness decreases with the increase in Inconel 718 content.

In addition, pores are also the main factor leading to hardness changes. In the area where the collapse occurs under the action of load and there are many pores, the ability of the specimen to suppress local deformation in the indentation process is poor, which leads to the low hardness in the corresponding area when the specimen is tested for hardness.

## 4. Conclusions

In this paper, the influences of different initial temperatures (room temperature and 300 °C) on the forming, microstructure morphology, and properties of Stellite-6/Inconel 718 FGM thin-walled parts were investigated, including macro morphology, element distribution, microstructure morphology, tensile properties, and microhardness. The following conclusions are drawn:High initial temperature can improve the uneven distribution of residual stress in the cladding layer by reducing the temperature gradient between the gradient layers, thus effectively reducing, or even eliminating, the crack sensitivity and avoiding the generation of cracks.Preheating will delay the solidification time of the molten pool, promote the uniform transition of interlayer elements, and help to avoid the sudden change in interlayer properties.Preheating can slow down or even inhibit the growth of columnar grains, which is beneficial to the nucleation of equiaxed grains. At the same time, the reduction in the temperature gradient makes the bonding area between the latter cladding layer and the former layer smaller, which increases the metallurgical bonding characteristics and improves the mechanical properties.The high initial temperature is conducive to the discharge of pores in the molten pool, avoiding the brittle fracture caused by local defects and reducing the mechanical properties. After preheating, the total area of pores at the fracture of FGM tensile specimen decreases, and the dimples are evenly and densely distributed. Compared with the FGM formed at room temperature, the mean yield strength, mean tensile strength, and mean elongation along the gradient direction are increased by 65.1 MPa, 97 MPa, and 5.2%, respectively.The high initial temperature has some influence on the hardness of FGM after forming, resulting in an average hardness of 26.9 HV less than that of FGM formed at room temperature.

## Figures and Tables

**Figure 1 materials-14-03609-f001:**
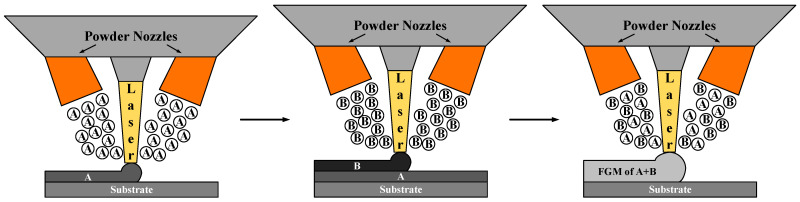
Principles of functional gradient functional material forming by coaxial powder feeding/laser fusion deposition technology.

**Figure 2 materials-14-03609-f002:**
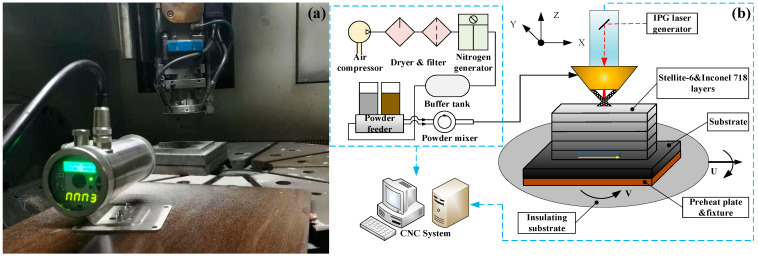
(**a**) SVW80C-3D hybrid additive and subtractive machine center; (**b**) principle of LMD process for Stellite-6/Inconel 718 FGM.

**Figure 3 materials-14-03609-f003:**
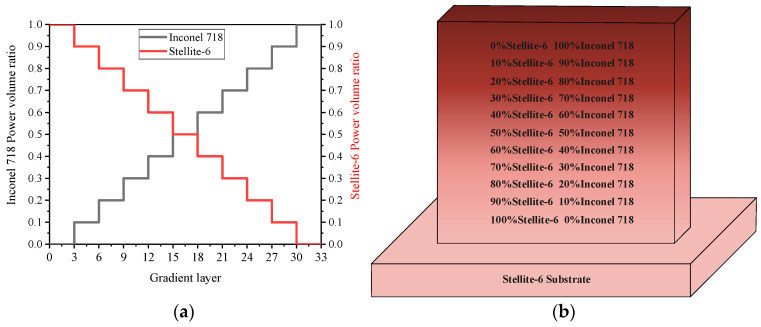
Stellite-6/Inconel 718 FGM: (**a**) powder volume ratio; (**b**) distribution of FGM layers.

**Figure 4 materials-14-03609-f004:**
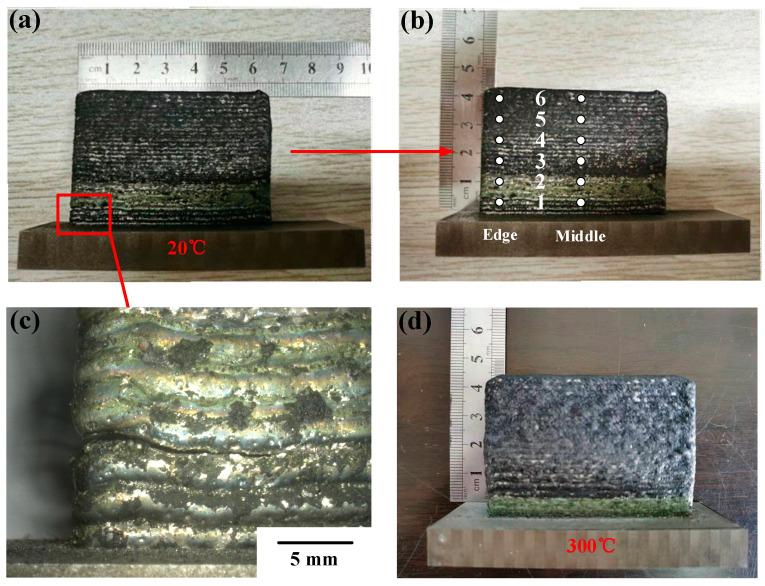
Macro morphology of the Stellite-6/Inconel 718 FGM thin-walled samples: (**a**) overall appearance (room temperature); (**b**) the measurement points located in different gradient areas; (**c**) partial crack areas; (**d**) overall appearance (300 °C).

**Figure 5 materials-14-03609-f005:**
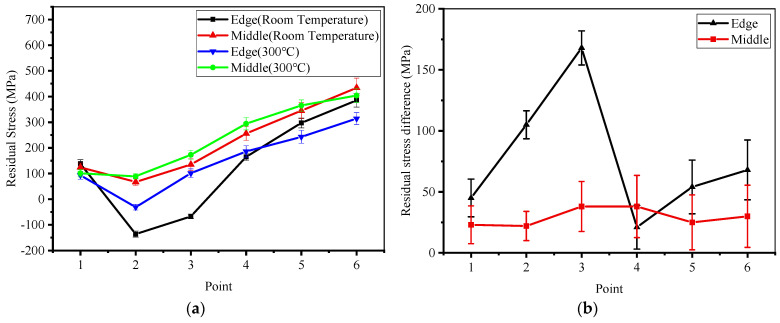
(**a**) Surface residual stress distribution of the FGM thin-walled samples; (**b**) residual stress difference at different height of the samples.

**Figure 6 materials-14-03609-f006:**
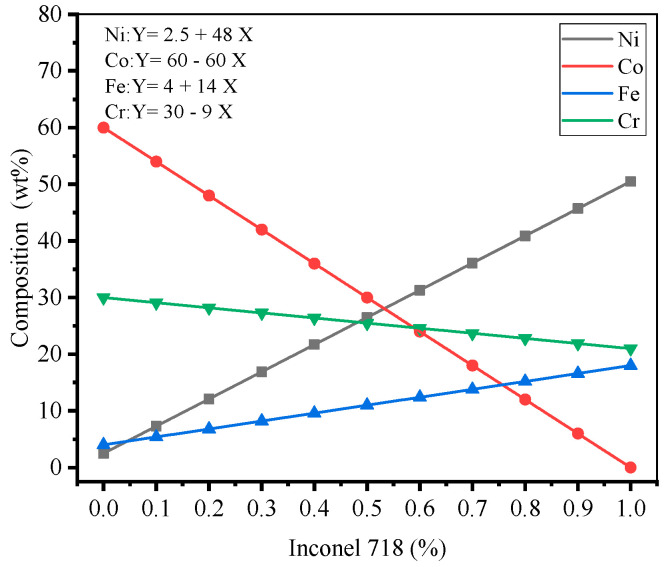
Design of Stellite-6/Inconel 718 FGM element distribution.

**Figure 7 materials-14-03609-f007:**
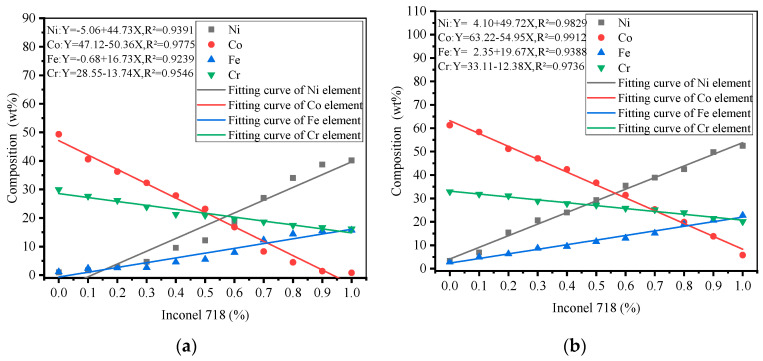
Element distribution of LMD forming FGM thin-walled parts: (**a**) 20 °C, (**b**) 300 °C.

**Figure 8 materials-14-03609-f008:**
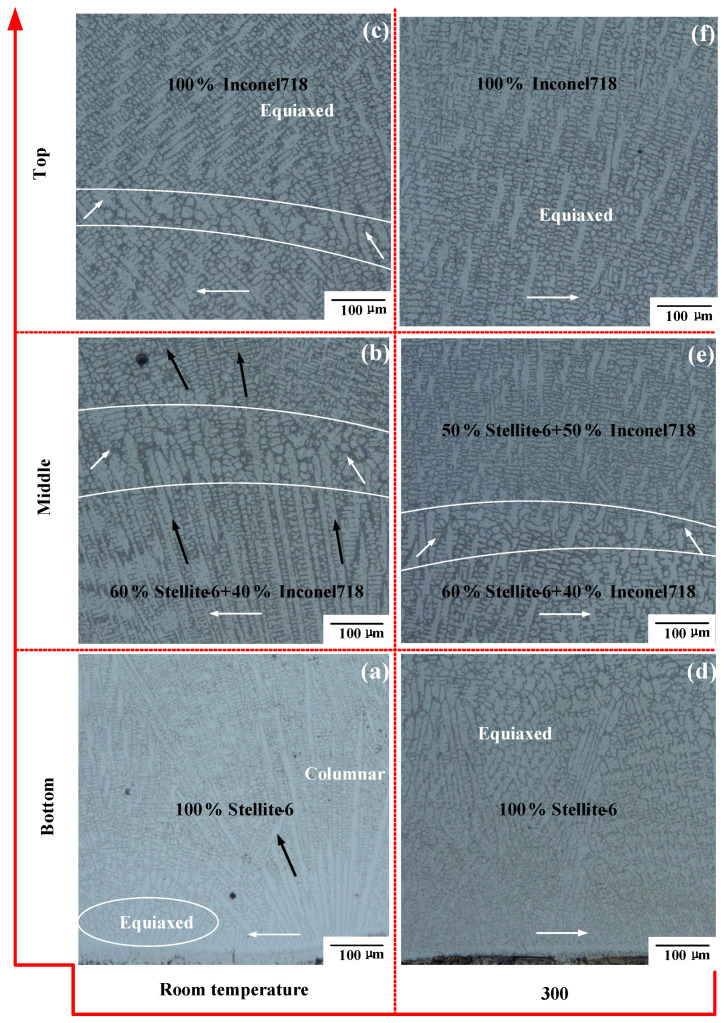
Dendritic morphology along the longitudinal section of the LMD FGM samples: (**a**) Bottom at room temperature, (**b**) Middle at room temperature, (**c**) Top at room temperature, (**d**) Bottom at 300 °C, (**e**) Middle at 300 °C, (**f**) Top at 300 °C.

**Figure 9 materials-14-03609-f009:**
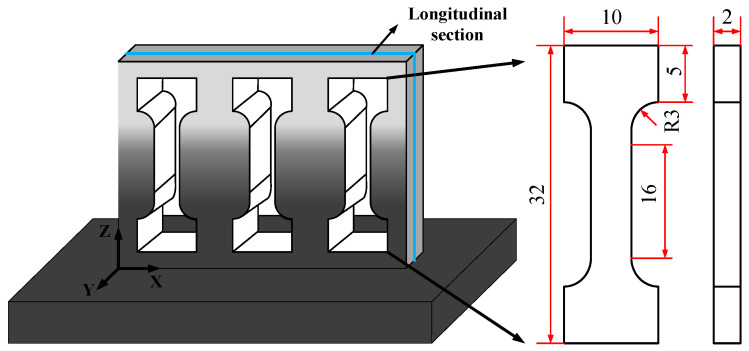
Position and geometry size (mm) of the tensile samples at the LMD thin-wall structure.

**Figure 10 materials-14-03609-f010:**
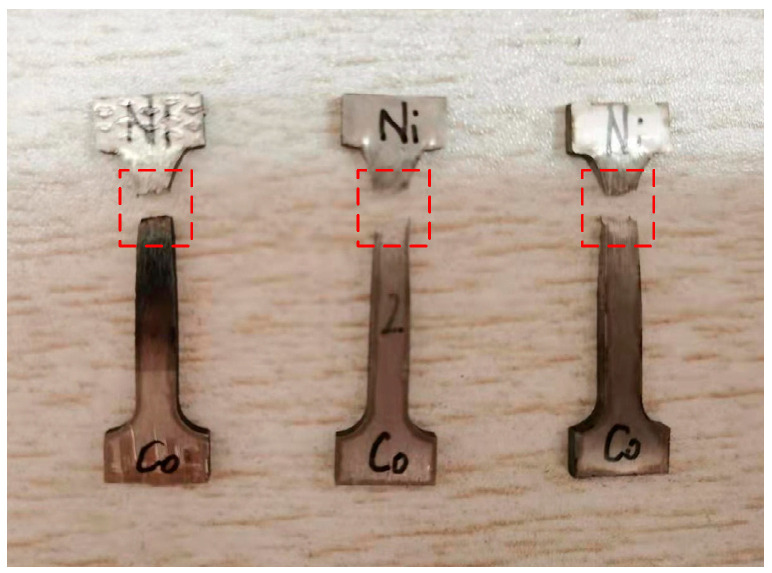
Fracture location of the tensile specimens.

**Figure 11 materials-14-03609-f011:**
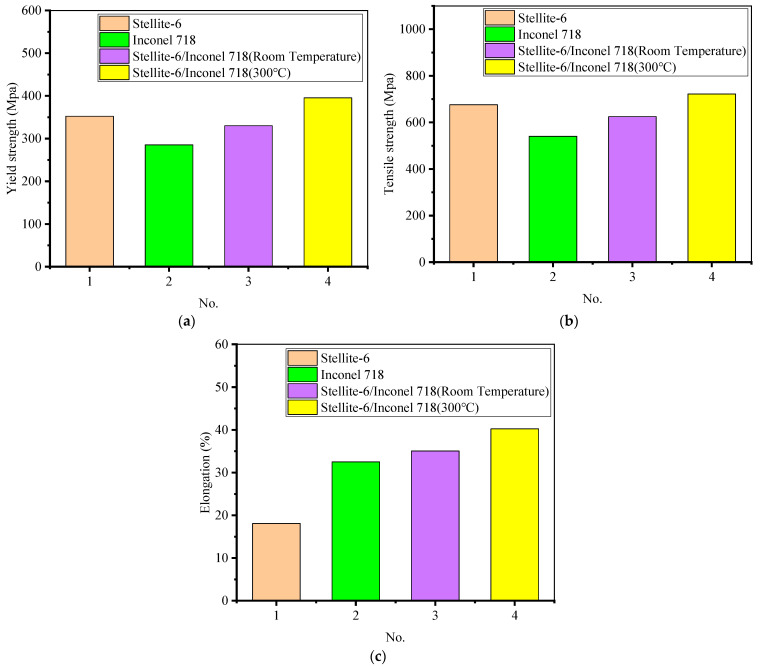
Tensile properties of Stellite-6/Inconel 718 FGM: (**a**) yield strength; (**b**) tensile strength; (**c**) elongation.

**Figure 12 materials-14-03609-f012:**
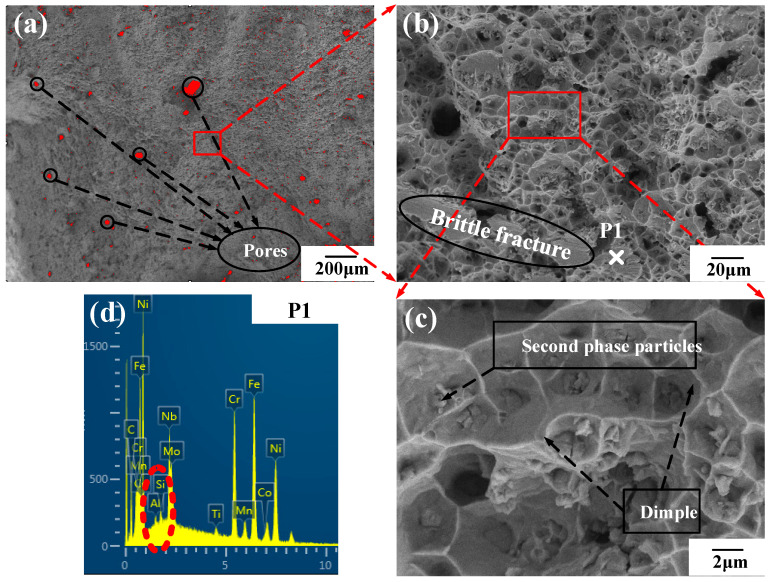
Microscopic fracture morphology and energy spectrum analysis of FGM tensile specimens at room temperature: (**a**) SEM 20×; (**b**) SEM 200×; (**c**) SEM 1000×; (**d**) test point P1 element content.

**Figure 13 materials-14-03609-f013:**
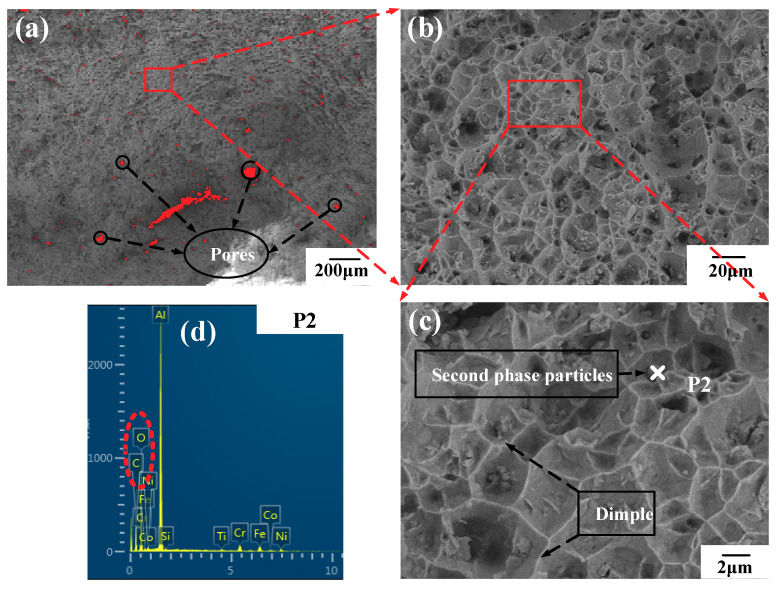
Microscopic fracture morphology and energy spectrum analysis of FGM tensile specimens at: 300 °C: (**a**) SEM 20×; (**b**) SEM 200×; (**c**) SEM 1000×; (**d**) test point P2 element content.

**Figure 14 materials-14-03609-f014:**
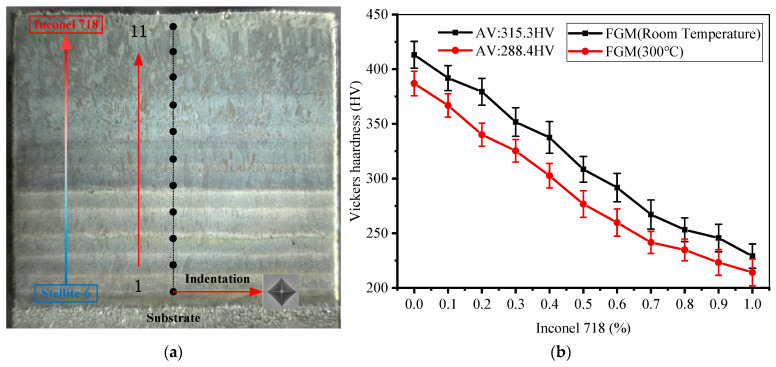
(**a**) Hardness test points on longitudinal section of FGM thin-walled parts; (**b**) Vickers hardness results at the test point.

**Table 1 materials-14-03609-t001:** Composition of Inconel 718 and Stellite-6 powder (wt.%).

Element	C	Si	Mn	Cr	Mo	Ti	Fe	Al	Co	Ni
Inconel 718	0.05	0.71	0.16	18.17	2.32	0.93	20.89	0.63	-	Bal.
Stellite-6	1.15	1.58	0.75	31.25	0.89	-	3.54	-	Bal.	2.54

**Table 2 materials-14-03609-t002:** Tensile properties of single material and FGM material.

Materials	YS/MPa	TS/MPa	EL/%
Stellite-6	352.5 ± 6.7	675.9 ± 5.4	18.10 ± 0.87
Inconel 718	285.4 ± 8.5	540.3 ± 7.1	32.50 ± 0.77
Stellite-6/Inconel 718 (20 °C)	330.2 ± 5.3	625.1 ± 8.4	35.05 ± 0.65
35.0Stellite-6/Inconel 718 (300 °C)	395.3 ± 9.6	722.1 ± 7.8	40.25 ± 0.85

## Data Availability

Not applicable.

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
