# Peer review of "Effect of Initial Temperature on the Microstructure and Properties of Stellite-6/Inconel 718 Functional Gradient Materials Formed by Laser Metal Deposition"

_materials, 2021, doi:10.3390/ma14133609_

Round 1
Reviewer 1 Report
It’s an interesting manuscript presenting the research concerning the influence of different initial temperatures on processing, microstructure morphology and properties of Stellite-6/Inconel 718 FGM thin-walled parts. The quality of the presented research results, drawings and discussions are at good level. There are, however, some imperfections that should be corrected:
- Table 1 header: not power, but powder….
- Figure 13 and Figure 14 captions should be improved – it should describe all pictures shown as a,b,c,d);
- The authors should consider showing the tensile tests results only by presenting Figure 12… Figure 11 is not necessary (and the figure caption is unclear) when all test data are shown in Figure 12 (this Figure should be, however, corrected because Figure 11 shows 6 flow stress curves).
Author Response
Detailed comments and response:
- Table 1 header: not power, but powder….
Response 1: According to the Reviewer, we have changed the misspelled word in Figure 1 from power to powder (page 2).
- Figure 13 and Figure 14 captions should be improved – it should describe all pictures shown as a,b,c,d). The authors should consider showing the tensile tests results only by presenting Figure 12… Figure 11 is not necessary (and the figure caption is unclear) when all test data are shown in Figure 12 (this Figure should be, however, corrected because Figure 11 shows 6 flow stress curves).
Response 2: According to the Reviewer, Figure 11 has been removed from the previous paper. The serial numbers of Figure 13 and Figure 14 are now Figure 12 and Figure 13. The titles of Figure 12 and Figure 13 have now been improved and (a), (b), (c) and (d) have been explained (page 11-12).

Reviewer 2 Report
Generally, article is properly written and has good scientific merit, however some minor mistakes have been detected:
- Figure 10 is missing.
- β parameter in table 2 is redundant, due to application of same treatment for all samples.
- Mechanical tests were performed at least 3 times for all samples, thus in table 2 the standard deviation of YS, TS and EL should be presented.
- EDS spectra are unreadable, should be bigger and limited to 10 keV.
Author Response
Detailed comments and response:
- Figure 10 is missing.
Response 1: According to the Reviewer, modified Figure 10 has been supplemented (page 9).
- β parameter in table 2 is redundant, due to application of same treatment for all samples.
Response 2: According to the Reviewer, the β has been removed (page 10).
- Mechanical tests were performed at least 3 times for all samples, thus in table 2 the standard deviation of YS, TS and EL should be presented.
Response 3: According to the Reviewer, the standard deviations of YS, TS and EL have been added to Table 2 (page 10).
- EDS spectra are unreadable, should be bigger and limited to 10 keV.
Response 4: According to the Reviewer, the EDS spectra have been modified (page 11-12).

Reviewer 3 Report
Manuscript number: materials-1268003
Title: Effect of Initial Temperature on the Microstructure and Properties of Stellite-6/Inconel 718 Functional Gradient Materials Formed by Laser Metal Deposition
This paper describes the influence of the initial temperature on the chemical composition, microstructure as well as selected mechanical properties during formed stellite-6/Inconel 718 FGM (Ni/Co-based alloy) by the LMD process. The manuscript is well written and results - interesting for application in technology. Please explain, why the Authors declare – the one value of the temperature as 20oC? In my opinion, more suitable will be a declaration of RT (room temperature). See also my other comments below:
The described research results depend on many technological and material factors, including the parameters of the LMD process (as mentioned by the Authors in the text) but also eg. porosity of the obtained FGM. Please specify (in section 2. Experimental Condition and Procedure) how many times have the tests been run (how many samples) have been tested for statistical analysis? What is the repeatability of such processes?
OTHER:
# Page 3, ABSTRACT: The first sentence must be deleted or correct.
# Pages 6, 10 and 13, Fig. 5, 12 and 15b: Please add the errors bars of presented data.
# Please use the suitable unit of selected mechanical parameters, i.e. MPa (don’t Mpa) in the whole manuscript (also on the Figs.)
Finally, I recommend the paper for publication in Materials but after minor revision.
Author Response
- This paper describes the influence of the initial temperature on the chemical composition, microstructure as well as selected mechanical properties during formed stellite-6/Inconel 718 FGM (Ni/Co-based alloy) by the LMD process. The manuscript is well written and results - interesting for application in technology. Please explain, why the Authors declare – the one value of the temperature as 20℃? In my opinion, more suitable will be a declaration of RT (room temperature). See also my other comments below:
Response: Thank you for your suggestion, the authors have found references in previous papers to comparative experimental studies of LMD performed at 20°C and 300°C preheated conditions. By referring to the research results of previous scholars, I propose in this paper a comparative analysis study of the performance of Stellite-6/Inconel 718 functional gradient materials formed using LMD at room temperature and preheated 300°C conditions. Changing 20°C to room temperature conditions is more accurate.
- The described research results depend on many technological and material factors, including the parameters of the LMD process (as mentioned by the Authors in the text) but also eg. porosity of the obtained FGM. Please specify (in section 2. Experimental Condition and Procedure) how many times have the tests been run (how many samples) have been tested for statistical analysis? What is the repeatability of such processes?
Response: According to the Reviewer, the Experimental Conditions and Procedure sections have been supplemented with the following: 10 groups of functional gradient material samples were formed.
- Page 3, ABSTRACT: The first sentence must be deleted or correct.
Response: According to the Reviewer, the first sentence in the abstract has been changed to: The versatility of the properties of the functional gradient material FGM varies with the composition and structure of the material, enabling formed parts to work properly in demanding environments such as ultra-high temperatures. To investigate……
- Pages 6 and 13, Fig. 5 and 14b: Please add the errors bars of presented data.
Response: According to the Reviewer, Figures 5 and 14b have been added to the data error bars.
- Please use the suitable unit of selected mechanical parameters, i.e. MPa (don’t Mpa) in the whole manuscript (also on the Figs.)
Response: According to the Reviewer, Mpa has been changed to MPa in the whole manuscript.
